# Smart System to Detect Painting Defects in Shipyards: Vision AI and a Deep-Learning Approach

Hanseok Ma [1] and Sunggeun Lee [2,*]

1    Smart Yard R&D Department, Daewoo Shipbuilding and Marine Engineering, 3370, Geoje-daero, Geoje-si 53302, Korea; mhs@dsme.co.kr
2    Division of Electronics and Electrical Information Engineering, Korea Maritime and Ocean University, 727, Taejong-ro, Yeongdo-gu, Busan 49112, Korea
*    Correspondence: sglee48@kmou.ac.kr

**Abstract:** The shipbuilding industry has recently had to address several problems, such as improving productivity and overcoming the limitations of existing worker-dependent defect-inspection systems for painting on large steel plates while meeting the demands for information and smart-factory systems for quality management. The target shipyard previously used human visual inspection and there was no system to manage defect frequency, type, or history. This is challenging because these defects can have different sizes, shapes, and locations. In addition, the shipyard environment is variable and limits the options for camera placements. To solve these problems, we developed a new Vision AI deep-learning system for detecting painting defects in an actual shipyard production line and conducted experiments to optimize and evaluate the performance. We then configured and installed the Vision AI system to control the actual shipyard production line through a programmable logic controller interface. The installed system analyzes images in real-time and is expected to improve productivity by 11% and reduce quality incidents by 2%. This is the first practical application of AI operating in conjunction with the control unit of the actual shipyard production line. The lessons learned here can be applied to other industrial systems.

**Keywords:** smart factory; deep learning; steel plate painting; painting defect detection; production line programmable logic controller

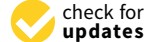

## 1. Introduction

The manufacturing industry values not only preparedness for manpower shortages but also responsibility in environmental, social, and governance (ESG) management. For this reason, companies are constantly investing in the construction of smart factories, vision systems, and smart manufacturing using artificial intelligence (AI).

Recently, most equipment related to manufacturing has been fully connected over a wireless network, monitored by sensors, and controlled by the manufacturing system. Computer intelligence reduces costs associated with the improvement of systems and product quality, thereby promoting productivity and sustainability. It provides various core technologies to cyber-physical systems (CPS) through the internet of things (IoT), cloud computing, and smart technologies. These technologies play a role in solving various problems while developing manufacturing technologies in modern society [1–3].

However, sufficient levels of stability and accuracy in production lines cannot be guaranteed by introducing such algorithms at many actual production sites including shipyards. The painting process line of the target shipyard is continuously monitored through CCTV at places that are difficult for humans to access. Such CCTV monitoring enables the determination of defects and the operation of the stop switch. It also completely stops the process line during work breaks or maintenance work. In addition, the absence of a quality management system decreases work efficiency and productivity. To overcome this, methods were proposed for process improvement in industrial sites using various methods.

In the existing research on defect detection methods, image data are acquired through a machine vision camera or 3D scanner in a controlled environment. Then, the system is studied to determine whether there is a defect using image processing using methods such as Bayesian classifiers, support vector machine classifiers [4], machine learning, or deep learning (DL) [5–7]. Image processing and machine learning techniques show high accuracy in image data that have fixed locations, shapes, or sizes. However, the accuracy decreases when there are environmental changes, such as the lighting, object size, or object shape. Most of the research studies could control the surrounding environment for the camera and lighting installation to achieve the optimal image data acquisition. However, when applied to an actual shipyard site, there are many restrictions on lighting and camera installations because the object is large and unstructured, and the surrounding equipment is huge. Therefore, a new system using DL was developed for use in industries such as shipbuilding that have unusual data that must be acquired under difficult constraints from the surrounding environments. This improved method was implemented by connecting to the control unit of the actual industrial equipment.

In this paper, we describe our novel system for the detection of painting defects in the upper and lower parts of steel plates in the process line. Our system incorporated deep learning with vision AI and demonstrably improved the production efficiency and quality of the painting process line by controlling the equipment with a programmable logic controller (PLC) interface.

The contributions of this study to the production site are summarized as follows:

1. This is an improved study configured to control the production line by applying deep learning at the actual shipyard production site.
2. It is relevant for conducting research using the unusual datasets acquired at other industrial shipyards.
3. Our system provided additional various services, such as history management and statistical analysis, which met the needs of users.

## 2. Related Methods

### 2.1. Current Methods

Most existing methods of quality inspection involve visual scrutiny by quality managers. In the case of the target factory, after the upper and lower parts of the steel plate were painted by the machine, a camera was installed and used to visually monitor the process of painting.

Figure 1 depicts a monitor-displayed a photograph of the processes occurring in the production site. This method not only fatigues workers but also stops the production line in the workers' absence. Therefore, other methods are needed to manage product quality and improve productivity.

### 2.2. Defect Detection Methods

#### 2.2.1. Handcrafted-Feature Method

Detection systems based on image processing have been developed and used to overcome the limitations of visual inspection. A representative example is a handcrafted-feature method that uses machine learning to define, beforehand, the feature points of defective images. It was applied to various industrial processes, such as those involving automobiles and semiconductors. This method is highly accurate when the input data are relatively standardized. However, when various types of input data or new patterns occur, its detection capabilities are limited [4].

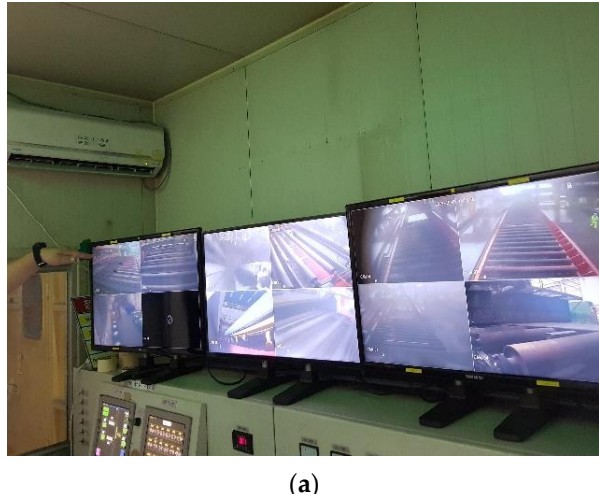 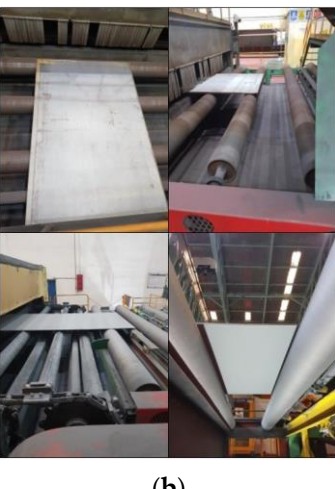

| (**a**) | (**b**) |

**Figure 1.** Painting factory site (**a**) CCTV monitors in operator room (**b**) Site.

### 2.2.2. Deep-Feature Method

In contrast to the handcrafted-feature method, the deep-feature method does not predefine feature points but allows computers to determine them from the data through a neural network.

The shipbuilding industry involves large amounts of order-taking and unstructured work. For this reason, it was difficult to automate and systematize activities such as processes and inspections. In particular, as mentioned in Section 2.2.1, the unstructured characteristics inherent in the input data of this industry make it difficult to apply the handcrafted-feature method. Unlike the handcrafted-feature method, the deep-feature method is expected to be more accurate in detecting defects in unstandardized steel plates of different types, including variations in seasons, ship types, etc. [5–7].

## 3. Deep-Learning Model for Detecting Painting Defects on Steel Plates

Figure 2 depicts a proposed method for detecting defects in actual steel plates using the deep-learning algorithm.

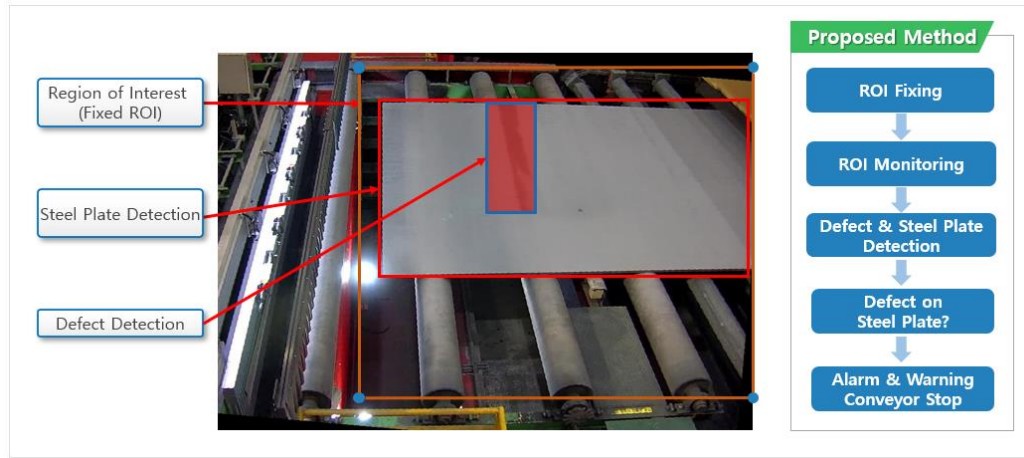

**Figure 2.** Proposed Method.

To overcome the limitations of visual inspection, it is essential to establish a system that can improve productivity and quality by using vision and deep-learning techniques in manufacturing industries such as the shipbuilding industry.

Therefore, we established a painting-defects detection system that can efficiently determine, record, manage, and control equipment with the aid of AI, whose superior

ability to perform simple repetitive tasks is of great advantage in making visual judgments in the environment of a shipyard.

### 3.1. Method Overview

As shown in Figure 2, the process steps for recognizing the defects in the steel plate are the following:

1.  The location on the conveyor at which the steel plate may be placed is fixed as the region of interest (ROI).
2.  Defect and steel plate classes are detected within the ROI.
3.  It is verified whether the class of the detected defect actually exists on the steel plate.
4.  If the answer to step 3 is in the affirmative, an alarm is sounded, and a warning is generated.

### 3.2. Camera and Lighting

Camera and lighting have the greatest influence on image data. In the case of the upper camera, acquiring data in a vertical direction is generally considered the best way. However, in the case of the actual shipbuilding factory, the height is 7 m. Additional structures could not be installed due to problems such as the maintenance of surrounding equipment. For this reason, the camera at the top had to be installed on the side wall, and a method of correcting the angle of the acquired image was used.

In the case of the bottom camera, in consideration of the distance between the conveyors and the absolute distance between the conveyor and the floor, it was installed at a position that can cover the imaging ROI when the largest steel plate is inserted.

In the case of the top lighting, the camera was installed using the lighting environment in the factory without installing additional lighting. In the case of the bottom lighting, only a specific part of the LED lamp was bright enough because of the angles. In the case of a large steel plate, a fluorescent lamp was selected.

Figure 3 shows the camera and lighting installation, and Table 1 lists the camera specifications.

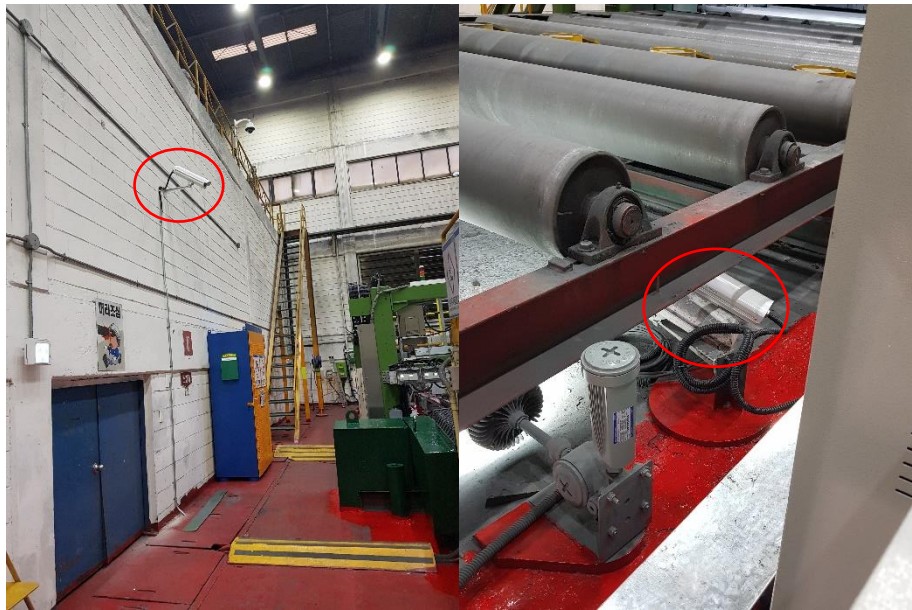

**Figure 3.** Camera and lighting installation.

**Table 1.** Camera specifications.

| Category | Specification |
| --- | --- |
| Image Device | 1/2.8" CMOS |
| Max. Resolution | 1920 × 1080 |
| Max. Frame rate | H.265/H.264: Max. 60 fps/50 fps (60 Hz/50 Hz) |
| Min. Illumination | Color: 0.05 Lux (F1.6, 1/30 s) BW: 0.005 Lux (IR LED On) |

### 3.3. Dataset Configuration and Explanation

The dataset classes and characteristics can be described as follows. The datasets collected for training were of two classes—normal and defective—whereas three classes were used in the deep-learning model—normal, defective, and dry. In winter, when the ambient temperature is very low, the painting sometimes does not dry out and it became difficult to distinguish such wet paint from the true defect pattern. Therefore, training data were acquired to represent this class of events, and the model was trained by classifying it as another class of data, labeled "Dry".

Figures 4–6 depict a set of images for each class with normal, defective, and "dry" painted plates that showed significantly similar patterns to defects.

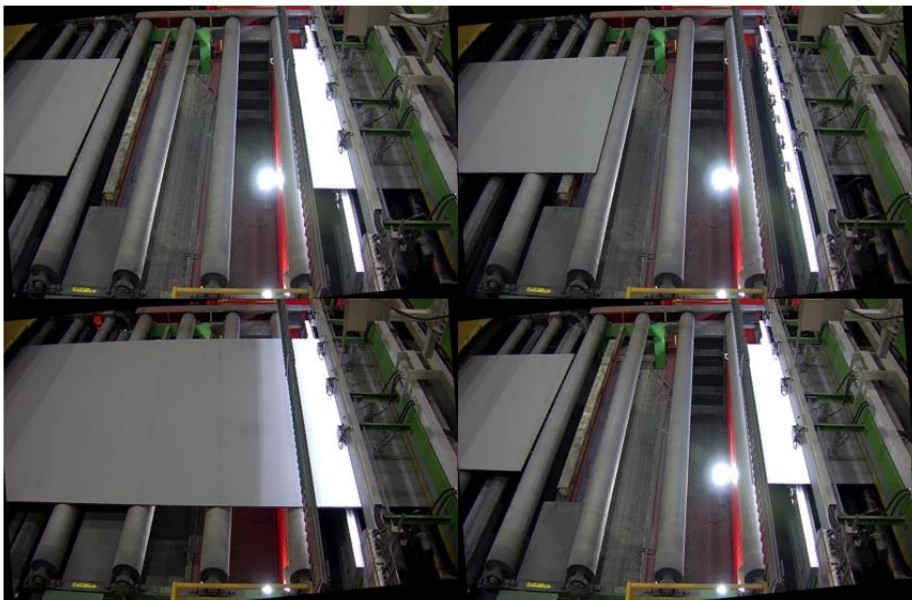

**Figure 4.** Sample images of normal steel plates.

A practical problem arose when the original dataset of the normal steel plates was sufficient, whereas those of the defective and dry steel plates were insufficient. Therefore, data augmentation was performed, and the final dataset consisted of approximately 4000 images.

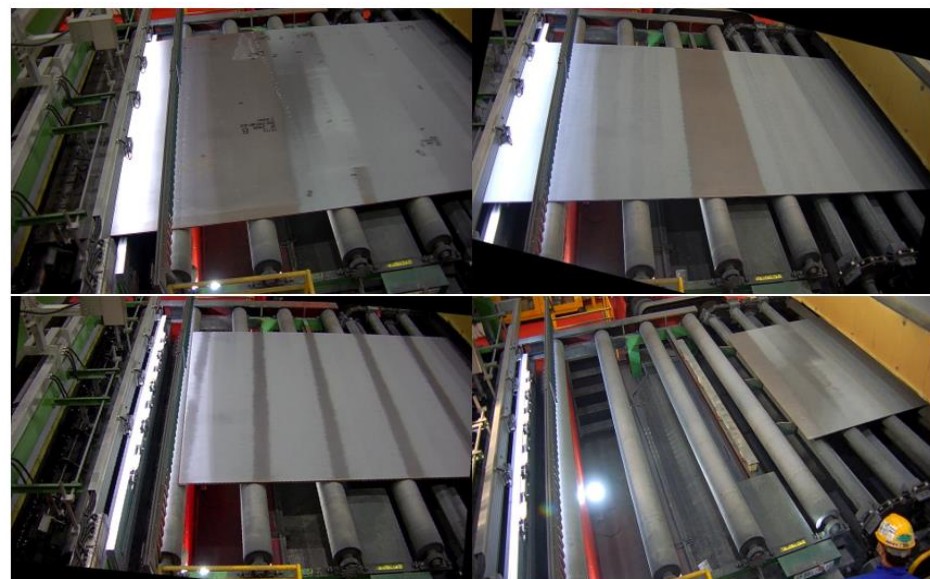

**Figure 5.** Sample images of defectively painted steel plates.

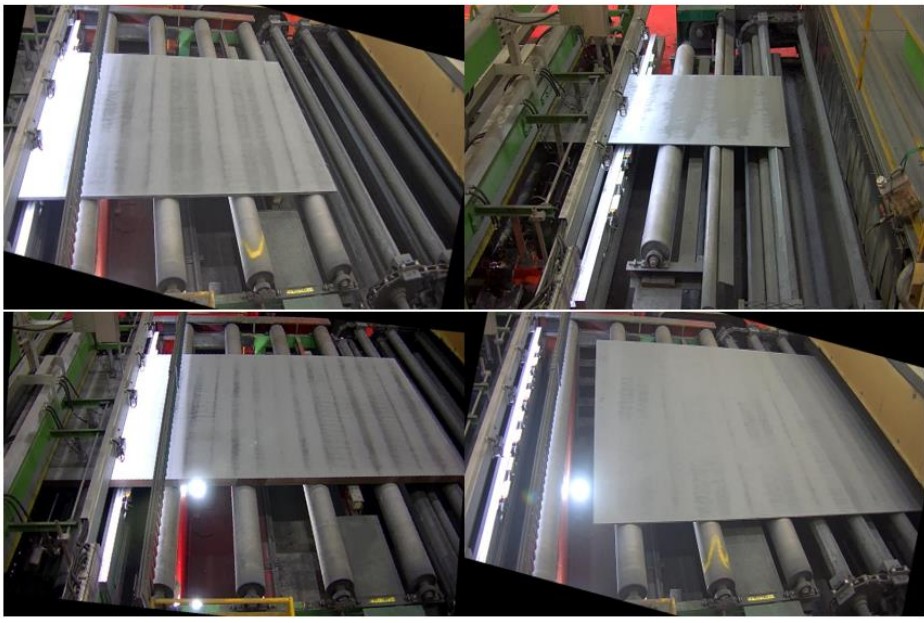

**Figure 6.** Sample images of painted, "dry" steel plates with wet paint.

*3.4. Experiments on Speed and Accuracy*

Since the production sites at the target factory were composed of fast-moving conveyor lines, swift judgment and action are important. Therefore, comparative experiments were conducted based on the you-only-look-once (YOLO) and single-shot multibox detector (SSD) models that are known for their high speed and accuracy.

In the case of YOLO, the model is lightweight and operates at a high speed, detecting defects in real-time. According to the paper *"YOLOv3: An Invention Improvement"*, a 28.2 mAP and 22 ms processing speed accuracy is nearly three times faster than the SSD model [8].

The SSD algorithm used in the comparative experiments used a single neural network to detect objects in an image. Unlike other detection methods that require an object proposal, the biggest advantage of the SSD is the collection of all operations in a single network, which eliminates the need to create proposals and re-sample pixels and their features. The SSD, trained with NVIDIA Titan X GPU to input images within the VOC2007 dataset,

displayed performance of 74.3 mAP and 59 FPS for images input with a resolution of 300 × 300 and 76.9 mAP for those inputs with a resolution of 512 × 512, performing better than the Fast R-CNN model [9]. The network of YOLO and SSD can be shown in Figure 7.

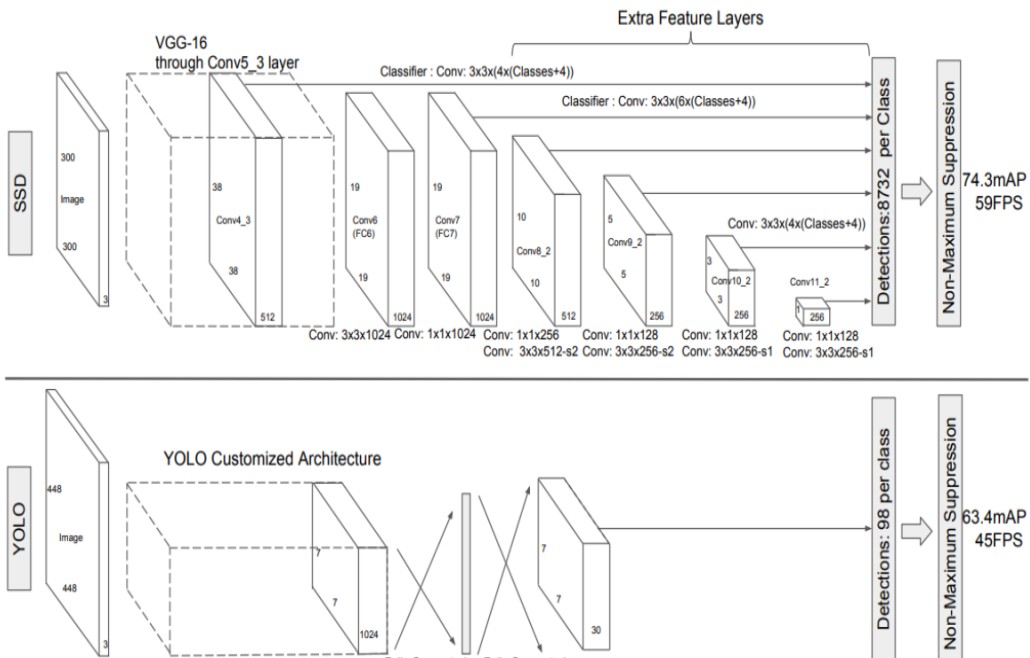

**Figure 7.** SSD and YOLO networks.

A comparative experiment was performed based on the data acquired by the camera at the top. This camera was selected for its superior ability to capture, over a wide range, the various types of data present at the site.

Table 2 shows the hyperparameters and Table 3 shows the hardware specifications used in the training and test experiments. Experiments 1 and 2 compared defect detection based on YOLO-v3 and SSD. Figures 8 and 9 show the YOLO-v3 mAP and loss graphs for 16,000 iterations and 200,000 iterations, respectively. Figure 10 shows the SSD mAP and loss graph.

**Table 2.** Hyperparameters used for training.

| Hyperparameter | YOLOv3 | SSD |
|---|---|---|
| Batch size | 64 | 64 |
| Learning rate | 0.00261 | 0.00261 |
| Input size | 416 × 416 | 300 × 300 |
| Number of steps | 200,000 | 200,000 |

**Table 3.** GPU server specifications.

| Category | Specification |
|---|---|
| GPUs | 4X Tesla V100 |
| TFLOPS | 500 |
| GPU Memory | 128 GB (total system) |
| Tensor Cores | 2560 |
| Cuda Cores | 20,480 |
| CPU | 2.2 GHz 20 cores |
| System Memory | 256 GB |

**Experiment 1. Defect-detection results based on YOLO-v3.**

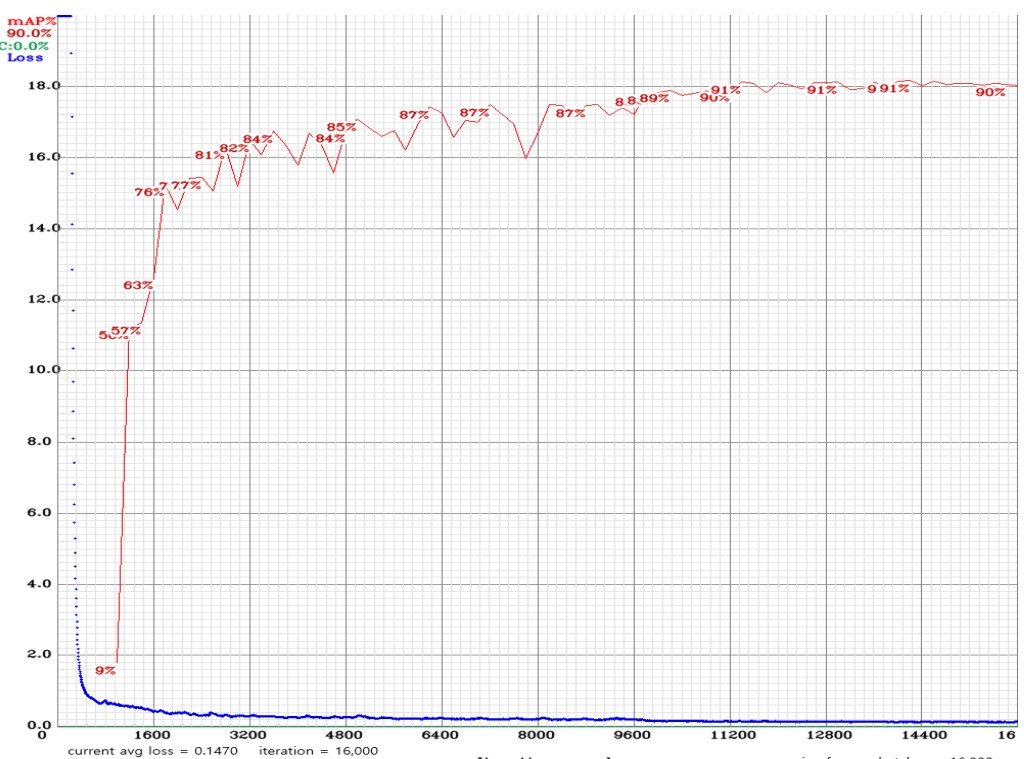

**Figure 8.** YOLO-v3 mAP and loss graph (16,000 iterations).

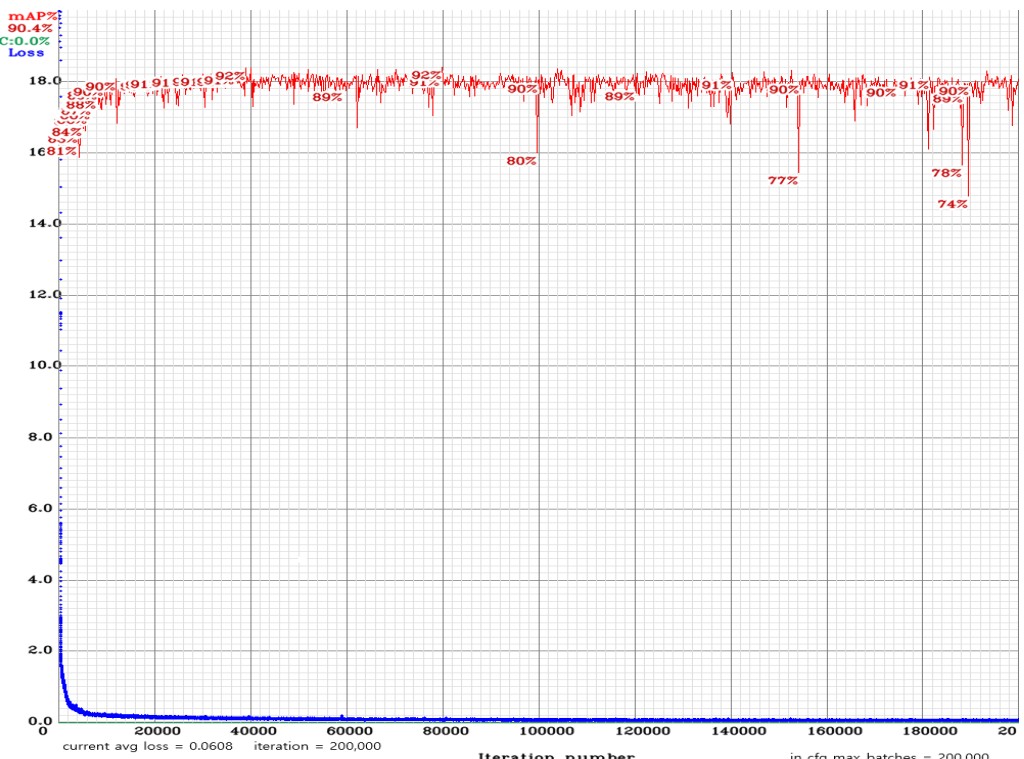

**Figure 9.** YOLO-v3 mAP and loss graph (200,000 iterations) of the top images.

The total dataset had 3532 images, of which 3179 (90%) were training and the remaining (353, 10%) were tests. The batch size at the time of training was 64; the training involved

approximately 12,800 iterations, corresponding to 256 epochs (1 epoch ≈ 50 iterations). Figure 8 shows the result of Experiment 1. The experiment converged to an average loss of 0.147. This was confirmed by training it for a minimum of 16,000 to a maximum of 200,000 iterations, as shown in Figures 8 and 9.

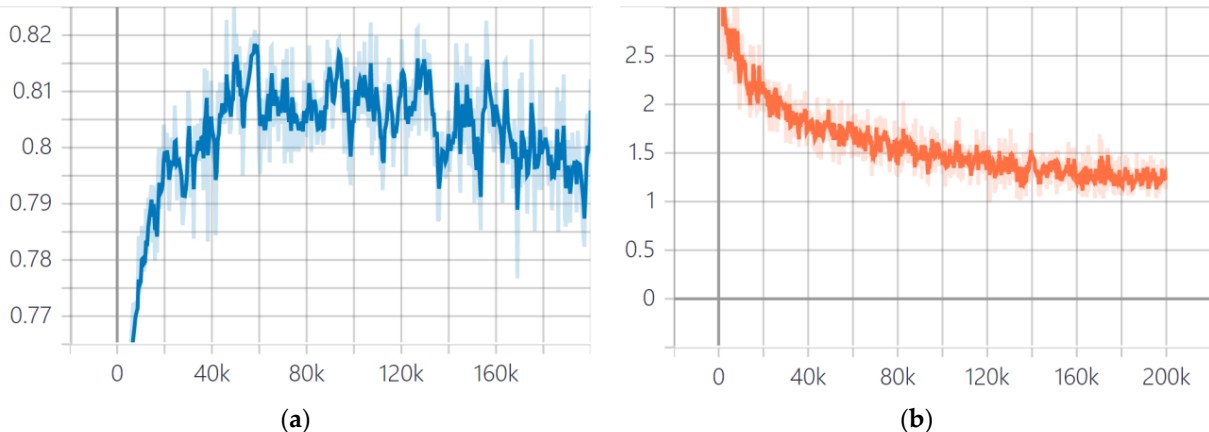

**Figure 10.** Experimental results of SSD (**a**) mAP (**b**) loss graph.

The loss corresponding to training with the YOLO-v3 model was approximately 0.06%, whereas the model itself had an accuracy of 90.4%.

**Experiment 2. Experimental results for SSD-based defect detection.**

The accuracy of the SSD model was 82%, and the loss was approximately 1%, as shown in Figure 10. Although the SSD processing speed, at 90 FPS, was similar to the YOLO-v3 model, the YOLO-v3 model was approximately 8% more accurate. Therefore, the YOLO-v3 model was selected for the final system development. Figure 11 shows the result of the top image. Thereafter, the YOLO-v3 model was also used to train the data acquired using the bottom camera.

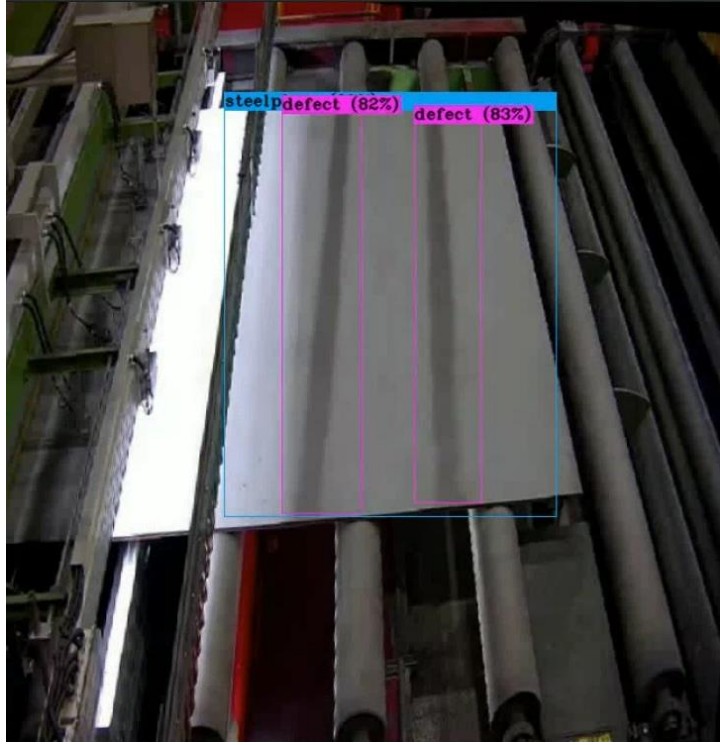

**Figure 11.** YOLO-v3-based result of top image.

Figure 12 shows the experimental results of the data acquired from the bottom of the plate. When compared to data obtained from the top, what was acquired from the bottom was inadequate and, due to the limited view angle, less accurate, with a rate of only 82% compared to the 90.4% accuracy from the top. However, the bottom data did exhibit sufficient defect-detection capability when it was actually operated in the shipyard system.

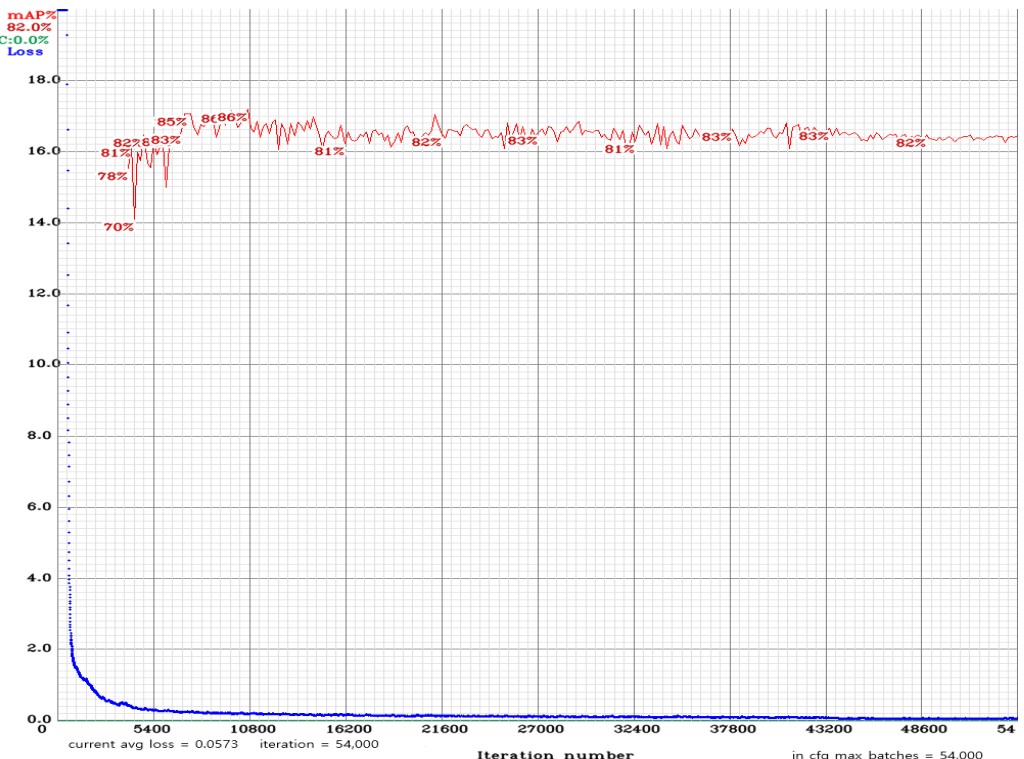

**Figure 12.** YOLO-v3 mAP and loss graph (200,000 iterations) of bottom images.

Experiments with the bottom data were conducted with varying parameters. It was determined that the maximum accuracy occurred at approximately 10,800 iterations (∼216 epochs). During the experiments, a detection error occurred due to incoming disturbances from the mining window at the top of the factory; nevertheless, these disturbances were filtered out and solved after converting the colors of the input image into HSV data [10].

In addition, a pattern resembling a defect appeared because of incomplete drying of the paint caused by low ambient temperatures. However, the system was retrained and adjusted for this. The corresponding data were solved by adding a "Dry" class to re-train the deep learning model, enabling it to classify such data as normal when processing events.

In the final configured system, HSV conversion was used in pre-processing, and the deep-learning model divided the data obtained from the steel plate into three classes—normal, error, and dry.

*3.5. Discussion*

In the experiments, the hyper-parameter was changed in various ways for each model, and filters were added or modified to the existing model. The performance of the basic network model was mostly high, and the accuracy varied from 85% to 90% for YOLO and 72% to 82% for SSD. In addition, at the beginning of data acquisition, fewer than 500 images of data were acquired for each line. Individual models showed higher accuracy than common models. However, building one model worked well on various defect patterns after more than 1000 images were acquired.

The strength of the system built using deep learning is that it can detect untrained defects. For example, during the actual development, a rusty steel plate was produced due to the omission of the blasting process. The DL model correctly judged it to be defective although this case was not in the training set. Although the patterns of defects trained in the actual field are similar, the rusty steel is a different type of defect. This detection of untrained defects is a strength of this system. Figure 13 shows the actual detection of the rusty steel plate as defective.

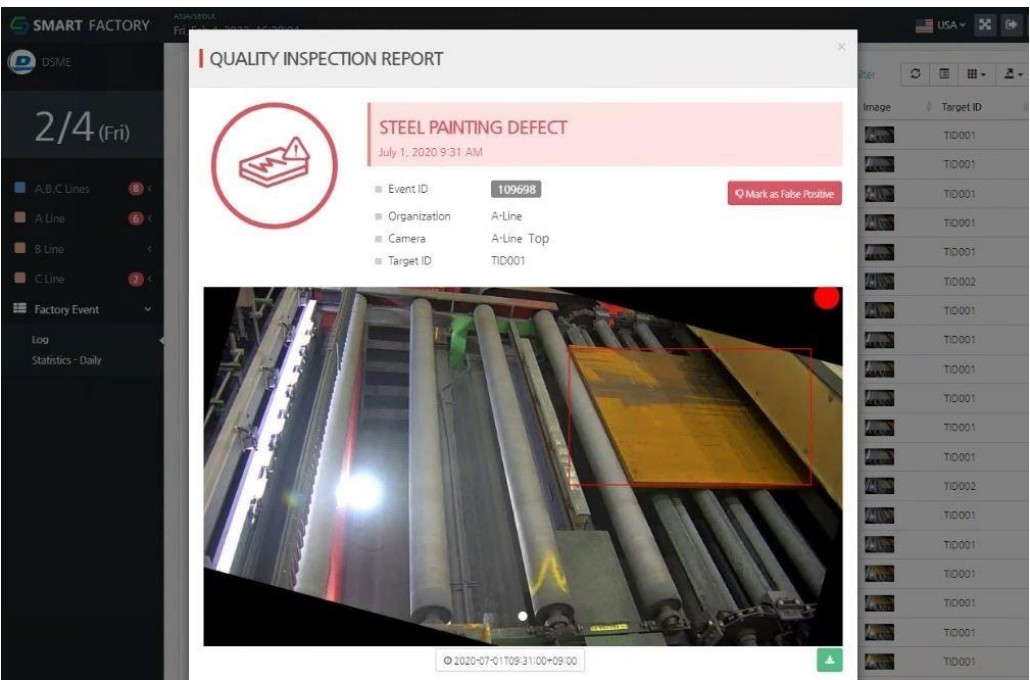

**Figure 13.** Detection of the rusty steel plate as defective.

Reliability is the most important factor in building an AI for field applications. To maintain this reliability, it is necessary to have a strong detection ability against unexpected, untrained errors.

The following points need to be considered for determining the limitations and maintenance of field systems that use the Vision AI.First, Ambient light should be controlled as much as possible, and the location of the camera and lighting affects the reliability of the system.Second, Since the location and angle of view of the camera may be changed due to the layout of the facility or an equipment change, it should be configured so that the trained model can operate universally by acquiring data from various angles during development.

## 4. Smart System for the Detection of Painting Defects

### 4.1. System Architecture

Figure 14 shows the configuration of the system, which consists of two cameras in each line, an image analysis server, a network video recorder (NVR) server, a web server, a database (DB), and an integrated control server.

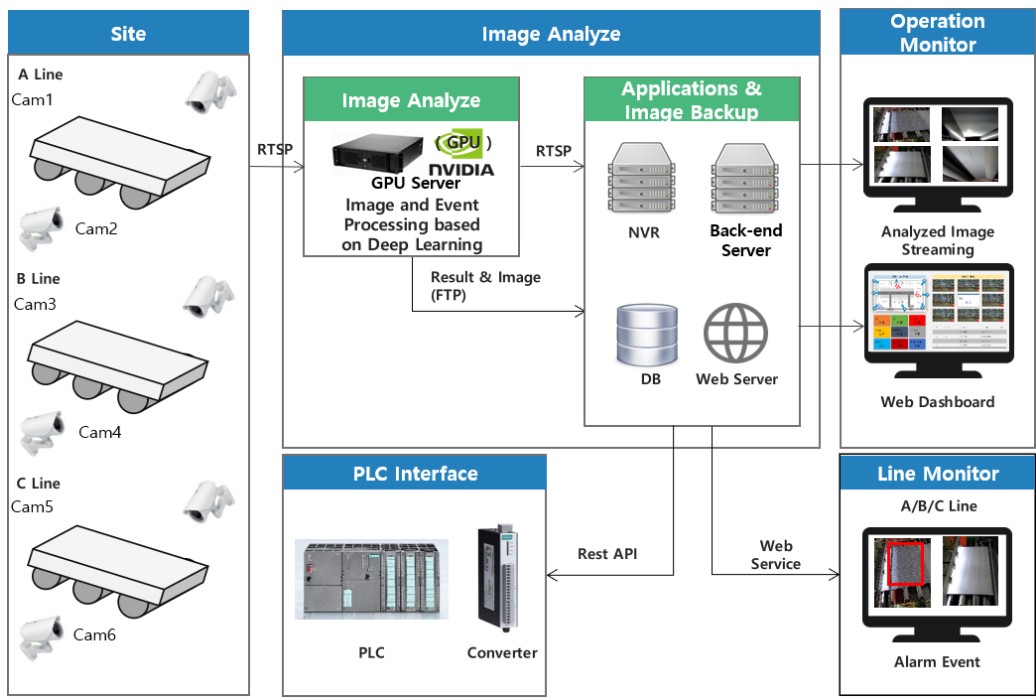

**Figure 14.** System configuration.

The images of each line, acquired via camera, were divided into normal and defective steel plates by the model that was trained through deep learning. The defect-detection algorithm recognized the steel plate and the defect as distinct objects in a fixed ROI when the acquired frame entered the input of the image analysis server as described in Section 3.1. It then determined whether the bounding box of the defect matched that of the steel plate.

However, since this system was configured to process at 15 FPS, defects in the same steel plate were repeatedly detected. To overcome this inaccuracy, an algorithm was developed to track the steel plate within the ROI while checking its upper left end. The flowchart of this algorithm is shown in Figure 15.

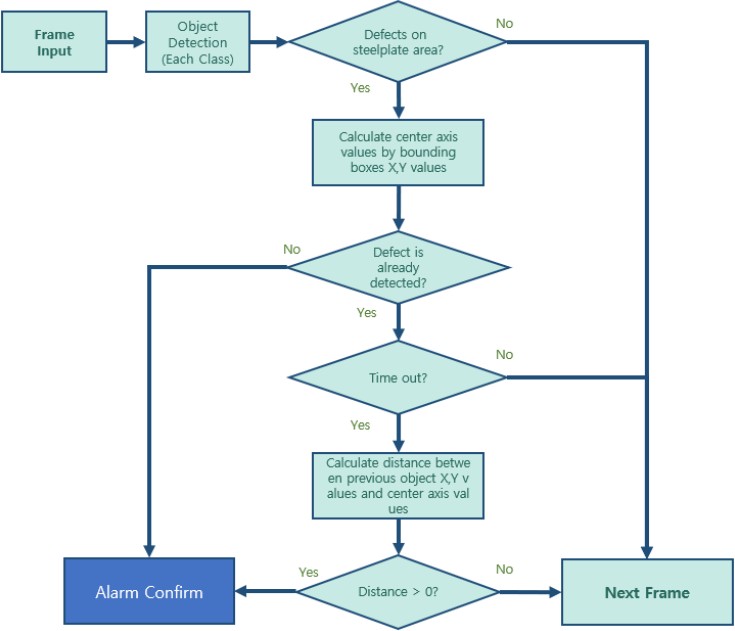

**Figure 15.** Flowchart of proposed algorithm.

### 4.2. Operating Applications

The system was configured to provide easy-to-use, high-quality services without unnecessary alarms or data accumulation when detecting defects during equipment operation.

Considering the shipyard environment, which already includes severe noise, our system is configured to alert a person to the detection of defects by not only recognizing them with pop-ups and sounding alarms but also by the interface through PLC by REST API to stop the conveyor. Figure 16 shows the hardware and site in which the actual final system was configured and implemented. Further details and the actual system are shown in Appendix A.

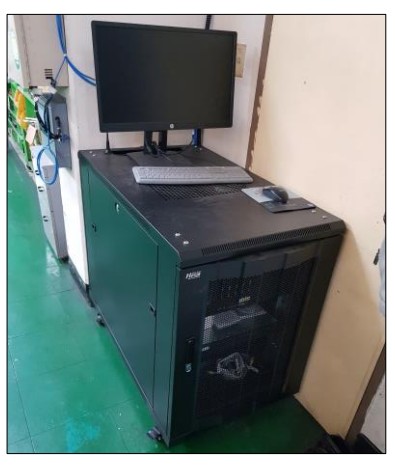

(**a**)

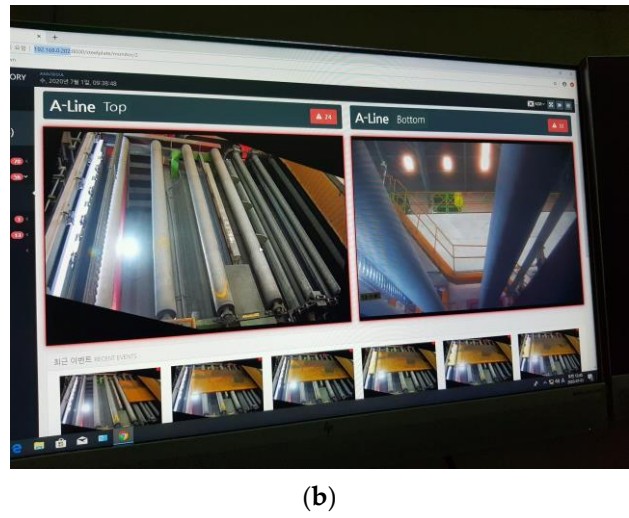

(**b**)

**Figure 16.** (**a**) Server hardware (**b**) Monitoring at the site.

### 5. Conclusions

If steel plates are moved to post-processing without detecting and resolving painting defects at an earlier stage, the quality and reliability of the final product are compromised, leading to a waste of time and capital resources. Hence, a system is necessary to detect painting defects at an earlier stage. We developed a DL-based defect-detection system and conducted experiments to evaluate the performance. The following conclusions were obtained as a result.

1. Before the implementation of this defect-detection system, monitoring workers frequently had to either stop the detection equipment if they were performing other maintenance tasks or needed to leave for a while. After incorporating this system, the defects were automatically detected, recorded, and controlled by reducing manual dependency, thereby reducing downtime by 2 h (approximately 11%), including 40 min. of rest and 80 min. of maintenance tasks. In addition, through history management, transparency in quality management was secured, thereby improving the reliability of the steel plates, which in turn boosted the reputation of ship owners and strengthened their competitiveness in winning orders.

2. When a painting defect occurs, sending the defective steel plate to the next process without rework or partial painting will lead to quality mishaps. Prior to implementing this new system, approximately 3% of such quality incidents occurred per quarter. This number was reduced to less than 1% after applying this system. In addition, by managing data, such as the images and frequencies of defects, the causes of these defects can be easily identified, which in turn helps to reduce defect incidence by helping with quick action and maintenance work.

3. During working hours, this new system greatly reduced the physical and psychological fatigue of workers who were constantly immersed in quality management work

through monitoring. Additionally, end-users of the product reported higher levels of satisfaction.

4. The previous work method was managed through CCTV for each line. In contrast, through the establishment of our integrated platform and service, the entire line could be monitored from anywhere, laying the foundation for responding to shortages of manpower.

5. Although various models of deep learning operate within generally known characteristics, in the case of industrial sites, the results of these models also vary with the object shapes and characteristics of the data. When the dataset is not standardized, existing models are selected based on known characteristics. However, it is advantageous to select as many models as possible and extensively experiment with them for fine-tuning.

6. The most challenging aspect of deploying the developed AI system is to form a positive change in people's perceptions. Although AI has recently become a prominent issue, AI has been regarded as a distant future story in many actual industrial sites. It is also true that there are many negative perceptions. Managers doubted the reliability of AI, and workers were afraid of losing their jobs. However, with the final developed AI system, it was recognized that AI can help workers rather than replace workers while improving productivity and quality with sufficient reliability.

The limitation of this study is that there are various types of defects, but sufficient data are not always secured for each defect, so they are classified as one class. Then the AI work is judged as defective based on repainting and specimen tests. In the future, the reliability and accuracy of the system can be improved by developing a model to accurately verify the location and shape of the defect using semantic or instance-segmentation techniques. In addition, we intend to study the effect of data deficiency on accuracy by applying data augmentation techniques using generative adversarial networks and will continue to optimize resources consumed through continuous model weight reduction using TensorRT algorithms. Furthermore, when a specific defect pattern occurs, its conditions will be continuously analyzed in connection with factors pertaining to the surrounding environment, such as temperature, humidity, pressure, and equipment speed. This information will also be used to manage and control the environment and equipment to improve quality.

**Author Contributions:** S.L. and H.M. conceived and designed the experiments; H.M. performed the experiments, software and visualization; S.L. analyzed and validated the data; S.L. and H.M. wrote the paper. All authors have read and agreed to the published version of the manuscript.

**Funding:** This research received no external funding.

**Institutional Review Board Statement:** Not applicable.

**Informed Consent Statement:** Not applicable.

**Data Availability Statement:** Not applicable.

**Conflicts of Interest:** The authors declare no conflict of interest.

# Appendix A

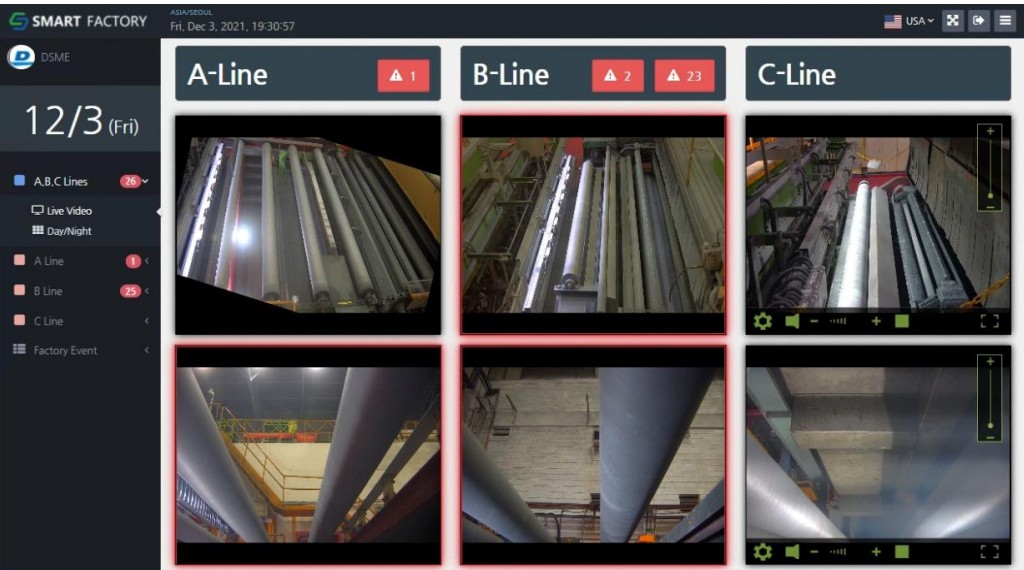

**Figure A1.** Live-streaming and monitoring.

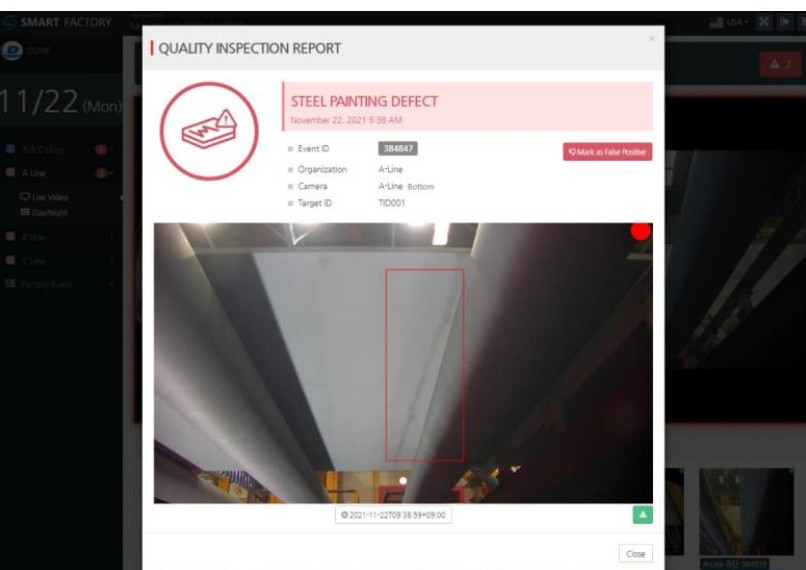

**Figure A2.** Defect-detection alarm and pop-up.

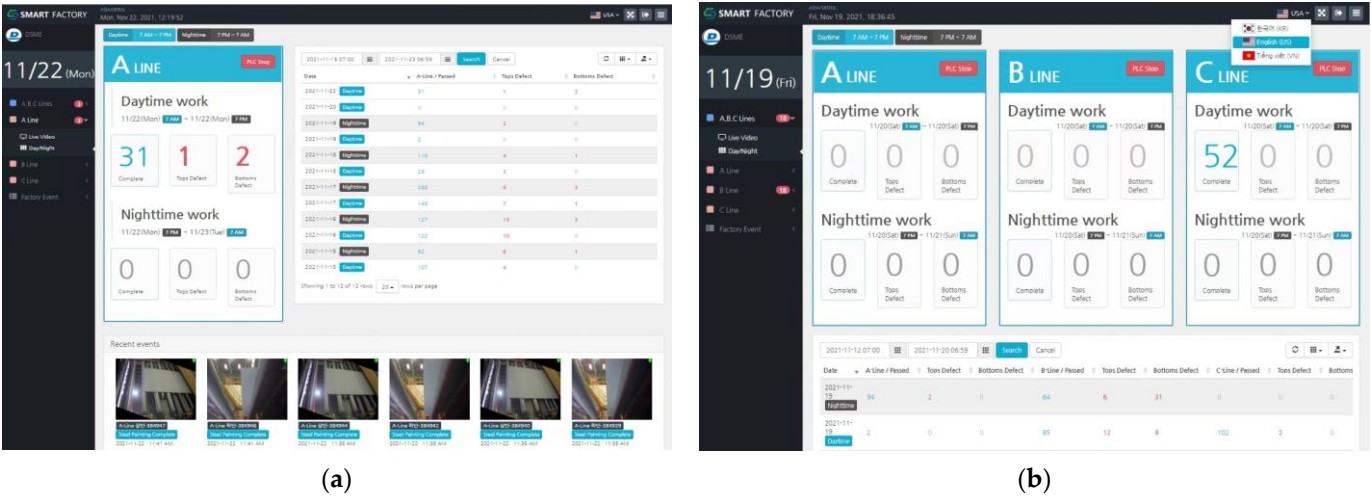

**Figure A3.** Summary and history management: (**a**) Each Line (**b**) Total.

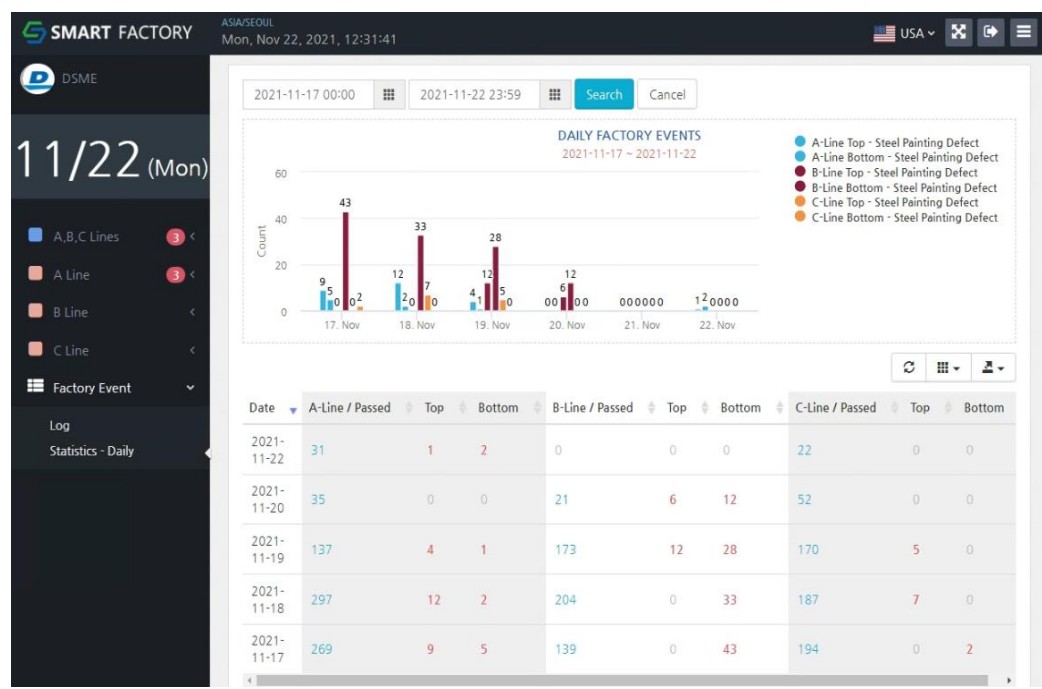

**Figure A4.** Log and statistics report.

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
