# Peer review of "Smart System to Detect Painting Defects in Shipyards: Vision AI and a Deep-Learning Approach"

_applsci, doi:10.3390/app12052412_

Round 1

Reviewer 1 Report

  1. In introduction missing the current research in the area inspection systems and defect detection
    1. Missing Issue definition  
    2. Focus on: "What is new?"
  2. In line 49, in my opinion, it is too bold statement. Maybe it is True. (Are you sure about it? What you do not know, usually automatically does not mean that it is not exist)
  3. In lines 72-74 is interesting statement, I suggest focusing on this issue: the possibility to capturing/evaluating untrained defects in inspection objects (defects of different shape, color, etc. from defects in dataset).
  4. How do you solve the light conditions and light from environments? Respectively, what is the impact of light conditions on detection accuracy?
  5. Is it possible to evaluate the detection accuracy in case of new color steel plate?
  6. What are the specifications of used camera devices?
  7. Why was used specific position camera  to inspect object? Were performed tests with different placements of the camera devices?

In my opinion, the current manuscript has many deficiencies. On the other hand, the manuscript after revising can provide great information of application AI (CNN, Supervised learning) and evaluate the limits of the current systems in case of use in practical industrial conditions. I advise focusing on edge condition and system possibilities in unknown conditions - occurring new types of defects, unknown colors of steel plates, (maybe different positions the camera devices, light condition, etc.) without retraining the model. 

Author Response

Dear reviewer in Applied Sciences

We are sincerely grateful for your thorough consideration and scrutiny of our manuscript, “Smart System to Detect Painting Defects in Shipyards: Vision AI, a Deep-Learning Approach,” Manuscript ID applsci-1583982. Through the accurate comments made by the reviewers, we better understand the critical issues in this paper. We have revised the manuscript according to the Reviewers’ suggestions. We hope that our revised manuscript will be considered and accepted for publication in the Applied Sciences. We acknowledge that the scientific quality of our manuscript was improved by the scrutinizing efforts of the reviewers and editors.

The changes within the revised manuscript are highlighted (underlined and in blue). Point-by-point responses to the reviewers’ comments are provided below.

Point 1: In introduction missing the current research in the area inspection systems and defect detection

  1. Missing Issue definition
  2. Focus on: "What is new?"

Response 1: We appreciate the reviewer’s comment.

Visual methods for the automated detection of paint defects in shipyards and similar environments are difficult because the product components and environments can be unstructured or amorphous, and the conventional methods applied to solve this problem are inadequate. The conventional methods such as computer vision, machine learning (ML) and deep learning (DL) have been applied to relatively standardized products, shapes, and locations. This paper was written to share our research and development results to overcome these limitations. Our study developed and applied a new system using deep learning (DL) that was developed with unusual data under the constraints of the variable shipbuilding surrounding environments. It was implemented by connecting it with the control unit of actual industrial equipment in a shipyard.

We have also added and revised this information to the revised Introduction section. The revised Introduction section is shown below:

“In the existing research on defect detection methods, image data is acquired through a machine vision camera or 3D scanner in a controlled environment. Then the system is studied to determine whether there is a defect using image processing using methods such as Bayesian classifiers, support vector machine classifiers[4], machine learning, or deep learning (DL) [5 7]. Image processing and machine learning techniques show high accuracy in image data that have fixed locations, shapes, or sizes. However, the accuracy decreases when there are environmental changes, such as the lighting, object size, or object shape. Most of the research studies could control the surrounding environment for the camera and lighting installation to achieve the optimal image data acquisition. However, when applied to an actual shipyard site, there are many restrictions on lighting and camera installations because the object is large and unstructured, and the surrounding equipment is huge. Therefore, a new system using DL was developed for use in industries like shipbuilding that have unusual data that must be acquired under difficult constraints from the surrounding environments. This improved method was implemented by connecting to the control unit of the actual industrial equipment.”

  1. Zhou, Q.; Chen, R. An automatic surface defect inspection system for automobiles using machine vision methods. Sensors 2019, 19(3), 644.
  2. Huang, D. C.; Lin, C. F. The Internet technology for defect detection system with deep learning method in smart factory. IEEE, Oxford, United Kingdom, 25th to 27th May 2018.
  3. Nguyen, H. T.; Yu, G. H. Defective Product Classification System for Smart Factory Based on Deep Learning. Electronics 2021, 10(7), 826.
  4. Chang, S. J.; Huang, C. Y. Deep Learning Model for the Inspection of Coffee Bean Defects. Applied Sciences 2021, 11(17), 8226.

Point 2: In line 49, in my opinion, it is too bold statement. Maybe it is True. (Are you sure about it? What you do not know, usually automatically does not mean that it is not exist)

Response 2: We appreciate the reviewer’s comment. We agree with your opinion, so we revised this sentence from “the first” to “improved”.

Here is the context for this study. DSME is a world-class company in the shipbuilding industry, and the three shipbuilding companies existing in Korea called as “Big 3” account for a large portion of the world's technology and production capacity about 25%. Big 3 are exchanging technologies while continuing to compete with each other in technology. The shipbuilding industry shares automation technology and applied production technologies to some extent.

There have been many studies that apply AI to design, procurement, and construction methods, but this is the first practical application of AI operating in conjunction with the control unit of the actual shipyard production line.

“1. This is an improved study configured to control the production line by applying deep learning at the actual shipyard production site.”

Point 3: In lines 72-74 is interesting statement, I suggest focusing on this issue: the possibility to capturing/evaluating untrained defects in inspection objects (defects of different shape, color, etc. from defects in dataset).

Response 3: We appreciate the reviewer’s comment. We wholly agree with the reviewer’s opinion.

When analyzing the data acquired to build this system, similar types of defects generally come out, but the defects found in the deployed system cannot be said to be the same shape. In this respect, it can be said that the biggest strength of this system built using Deep Learning is that defects in various shapes and locations can be detected rather than being restricted. For example, during the actual development, a rusty steel plate was detected due to the omission of the blasting  process, which was judged to be defective. This content was added to the content of the paper.

Considering these points, future studies aim to more accurately detect the shape and location of defects, analyze the causes of them, and further acquire data from surrounding environments and equipment to build a system that can prevent defects.

“The strength of the system built using deep learning is that it can detect untrained defects. During the actual development, a rusty steel plate was detected due to the omission of the blasting process, which was judged to be defective. In addition, although the patterns of defects trained in the actual field are similar, this is not exactly the same type of defect, so this can be said to be the strength of this system. Figure 13 shows the actual detection of the rusty steel plate as defective.”

Figure 13. Detection of the rusty steel plate as defective

Point 4: How do you solve the light conditions and light from environments? Respectively, what is the impact of light conditions on detection accuracy?

Response 4: In the case of the top lighting, the camera was installed in consideration of the lighting environment in the factory without installing additional lighting, and in the case of the bottom, only a specific part of the LED lamp was bright due to straightforwardness, and in the case of a large steel plate, the fluorescent lamp was selected.

In the case of environmental lighting, as mentioned 3.4 line 192 – 193 in the paper, shadows caused by disturbances and disturbances occurred through mining windows, and if data were insufficient, the shadow itself was recognized as defective. However, this problem cannot be considered to have been 100% resolved. When upgrading the system in the future, it is planned to physically block it to minimize the impact of external lighting as much as possible. Specific methods are considering blocking the upper window that affects the location with cell cardboard such as black and yellow, and installing blackout curtains around the location.

Point 5: Is it possible to evaluate the detection accuracy in case of new color steel plate?

Response 5: Since the target factory is painted by similar steel types and predetermined paint, there was no opportunity for judging the accuracy of a new color. Therefore, the test was not performed. However, if there is a process of blasting before the painting process. If this process does not proceed normally, a rusty steel plate may come in, which was detected as a defect as mentioned above.

If there is an opportunity, we would like to have an opportunity to verify that the system can be applied to similar factories to verify that the constructed systems and models can be used universally.

Point 6: What are the specifications of used camera devices?

Response 6: We appreciate the reviewer’s comment. We missed describing camera specifications. It is additionally tabulated as below:

“Table 1 is a camera specification.”

Table 1. Camera specifications

Category

Specification

Image Device

1/2.8" CMOS

Max. Resolution

1920x1080

Max. Framerate

H.265/H.264: Max. 60fps/50fps(60Hz/50Hz)

Min. Illumination

Color: 0.05Lux(F1.6, 1/30sec) BW: 0.005Lux(IR LED On)

Point 7: Why was used specific position camera to inspect object? Were performed tests with different placements of the camera devices?

Response 7: Tests were conducted to install cameras in various fields.

Ideally, both the upper and lower parts would be measured in the vertical direction relative to the object. However,, they were most affected by the surrounding environment and installable locations. The positions were selected based on obtaining the best image among the installable locations.

In the case of the top, it was difficult to install the vertical upper part over 7 m in height, and it was difficult to install additional structures on the upper part due to problems such as maintenance of factory equipment. In conclusion, it was inevitable to install it on the side wall. After that, a place was selected where the influence on the surrounding lighting environment and the covered structure could be minimized.

There was a much more difficult problem at the bottom. With the actual width of the steel plate reaching up to 4.5 m, the distance between the conveyor rollers was about 1.5 m maximum, and the absolute distance between the roller and the floor was 0.6 m or less, which inevitably caused spatial restrictions on installing the camera. Therefore, in the case of the bottom, the position where the entire image can be obtained was the highest priority, and the position of the light was adjusted based on this. The test was conducted in two places, and the quality of the image was confirmed.

 Related items were additionally added as 3.2 Camera and Light as below:

“Camera and lighting have the greatest influence on image data. In the case of the upper camera, acquiring data in a vertical direction is generally considered the best way. However, in the case of the actual shipbuilding factory, the height is 7 m. Additional structures could not be installed due to problems such as the maintenance of surrounding equipment. For this reason, the camera at the top had to be installed on the side wall, and a method of correcting the angle of the acquired image was used.

In the case of the bottom camera, in consideration of the distance between the conveyors and the absolute distance between the conveyor and the floor, it was installed at a position that can cover the imaging ROI when the largest steel plate is inserted.

In the case of the top lighting, the camera was installed using the lighting environment in the factory without installing additional lighting. In the case of the bottom lighting, only a specific part of the LED lamp was bright enough because of the angles. In the case of a large steel plate, a fluorescent lamp was selected.

Figure 3 shows the camera and lighting installation, and Table 1 lists the camera specifications.”

Figure 3. Camera and lighting installation

GENERAL COMMENTS: In my opinion, the current manuscript has many deficiencies. On the other hand, the manuscript after revising can provide great information of application AI (CNN, Supervised learning) and evaluate the limits of the current systems in case of use in practical industrial conditions. I advise focusing on edge condition and system possibilities in unknown conditions - occurring new types of defects, unknown colors of steel plates, (maybe different positions the camera devices, light condition, etc.) without retraining the model.

Response : We appreciate the reviewer’s comment. We wholly agree with the reviewer’s opinion. Although the core of this system is to build a defect detection system using AI, the purpose of this paper is to share various experiences and trials and errors experienced while applying this system to the actual site. Although deep learning was used, it was thought that sharing these experiences, which were not new or significant in terms of algorithms, was in line with the purpose of the journal Applied Sciences, while sharing many experiences in AI applicability research in the future. We did evaluate the occurrence of new types of defects, unknown color (rusting) of steel plates and lighting factors without retraining the model. Our revised manuscript will clarify this.

Reviewer 2 Report

The paper is well written. The developed methodology and its are application are of interest the scientific community as, despite the the CV based deep learning algorithm is not novel, it is applied to real data and currently deployed (or at least it is in test stage).

Before publication, I suggest the Authors to tackle the following (minor) issues:

1) provide a more detailed literature review on similar applications;

2) add more details on how all the different neural networks hyperparameters have been set;

3) discuss which were the most changelling aspects of deploying the developed AI  system;

4) discuss which are the main challenging of maintining the developed AI detection system.

Author Response

Dear reviewer in Applied Sciences

We are sincerely grateful for your thorough consideration and scrutiny of our manuscript, “Smart System to Detect Painting Defects in Shipyards: Vision AI, a Deep-Learning Approach,” Manuscript ID applsci-1583982. Through the accurate comments made by the reviewers, we better understand the critical issues in this paper. We have revised the manuscript according to the Reviewers’ suggestions. We hope that our revised manuscript will be considered and accepted for publication in the Applied Sciences. We acknowledge that the scientific quality of our manuscript was improved by the scrutinizing efforts of the reviewers and editors.

The changes within the revised manuscript are highlighted (underlined and in blue). Point-by-point responses to the reviewers’ comments are provided below.

GENERAL COMMENTS: The paper is well written. The developed methodology and it’s are application are of interest the scientific community as, despite the the CV based deep learning algorithm is not novel, it is applied to real data and currently deployed (or at least it is in test stage).

Response : We appreciate the reviewer’s comment. We wholly agree with the reviewer’s opinion and thank you for understanding the exact purpose of this paper. Although algorithm research is important, we believe that the difficulties experienced in applying it to actual practical industrial sites and the characteristics of existing algorithms for real data are also meaningful to the readers of Applied Sciences.

When we trained industrial field data with various algorithms while building this system, there were issues such as different accuracy from general datasets like MSCOCO. These issues will also be shared in future studies through more specific and diverse experiments.

Point 1: provide a more detailed literature review on similar applications

Response 1: We appreciate the reviewer’s comment. In the most similar areas, many studies have been conducted on painting and surface quality in the semiconductor and automobile fields. This was the first time in the field of shipbuilding. There were similar studies in the field of steel manufacturing but has been no other study in the complex shipyard type of environment.

In the method of visual inspection of most workers, there have been many papers about inspection methods based on computer vision, machine learning, and deep learning.

We added revised material about similar applications as below:We have also added and revised this information to the revised Introduction section. The revised Introduction section is shown below:

“In the existing research on defect detection methods, image data is acquired through a machine vision camera or 3D scanner in a controlled environment. Then the system is studied to determine whether there is a defect using image processing using methods such as Bayesian classifiers, support vector machine classifiers[4], machine learning, or deep learning (DL) [5 7]. Image processing and machine learning techniques show high accuracy in image data that have fixed locations, shapes, or sizes. However, the accuracy decreases when there are environmental changes, such as the lighting, object size, or object shape. Most of the research studies could control the surrounding environment for the camera and lighting installation to achieve the optimal image data acquisition. However, when applied to an actual shipyard site, there are many restrictions on lighting and camera installations because the object is large and unstructured, and the surrounding equipment is huge. Therefore, a new system using DL was developed for use in industries like shipbuilding that have unusual data that must be acquired under difficult constraints from the surrounding environments. This improved method was implemented by connecting to the control unit of the actual industrial equipment.”

  1. Zhou, Q.; Chen, R. An automatic surface defect inspection system for automobiles using machine vision methods. Sensors 2019, 19(3), 644.
  2. Huang, D. C.; Lin, C. F. The Internet technology for defect detection system with deep learning method in smart factory. IEEE, Oxford, United Kingdom, 25th to 27th May 2018.
  3. Nguyen, H. T.; Yu, G. H. Defective Product Classification System for Smart Factory Based on Deep Learning. Electronics 2021, 10(7), 826.
  4. Chang, S. J.; Huang, C. Y. Deep Learning Model for the Inspection of Coffee Bean Defects. Applied Sciences 2021, 11(17), 8226.

Point 2: add more details on how all the different neural networks hyperparameters have been set

Response 2: We appreciate the reviewer’s comment. We revised the manuscript to add more details. Hyperparameters were tabulated for each algorithm as below:

“Table 2 shows the hyperparameters”

Table 2. Hyperparameters used for training

Hyperparameter

YOLOv3

SSD

Batch size

64

64

Learning rate

0.00261

0.00261

Input size

416 x 416

300 x 300

Number of steps

200,000

200,000

Point 3: discuss which were the most changelling aspects of deploying the developed AI system

Response 3: Deploying the systems required a positive change in people's perception. Although AI has recently become a big issue in general, in actual industrial sites AI has been regarded as a story about the distant future story. It is also true that there are many negative perceptions. Managers doubted the reliability of AI, and workers were afraid of losing their jobs. However, with the developed AI system, it was recognized that AI can help workers make more contributions to the production site, not replace workers, while it contributes to productivity and quality with sufficient reliability.

We think it's a very good point, and we have added these in Conclusions as below:

“The most changing aspects of deploying the developed AI system is a positive change in people's perception. Although AI has recently become a big issue, AI has been regarded as a distant future story in actual industrial sites. It is also true that there are many negative perceptions. Managers doubted the reliability of AI, and workers were afraid of losing their jobs. However, with the developed AI system, it is recognized that AI can help workers and contribute a lot to the production site, not to replace workers, as it contributes to productivity and quality with sufficient reliability.”

Point 4: discuss which are the main challenging of maintining the developed AI detection system.

Response 4: We appreciate the reviewer’s comment.

There are two points to consider in terms of system and hardware.

  1. In the case of AI models, it is still difficult to apply universally, so continuous updates are needed.

In fact, in the particular case of the bottom of the C Line in the shipyard, the camera and lighting needed to be moved due to surrounding problems. Although some defects were detected, the accuracy was relatively reduced, so additional data acquisition and learning were required. As these cases may continue to appear in the future, it is important to acquire and train data of various angles and shapes to become a model and system that are less affected by environmental changes as much as possible.

  1. There is not only a painting process in the factory, but also a blasting process, and there is a workshop such as paint dilution around it, so dust accumulates in cameras. Of course, there are covers to prevent this, but if direct contaminants are stained on lenses, the image itself may be distorted, so continuous maintenance is required. Fortunately, although the system currently applied has been operating for a considerable amount of time, it is being managed well. In the long run, it is deemed that good management is essential because it would have a direct impact on the system reliability of the system.

We mentioned systematic maintenance in the paper as follows:

“In other words, the following points to be considered for the limitations and maintenance of the field system applied with Vision AI are as follows.

  1. Ambient light should be controlled as much as possible, and the location of the camera and lighting affects the reliability of the system.
  2. Since the location and angle of view of the camera may be changed due to layout of the facility or equipment change, it should be configured so that the trained model can operate universally by acquiring data from various angles during development.”

Round 2

Reviewer 1 Report

Thanks to authors for answering my questions. First, I appreciate the deployment inspection system in the real Industry condition. My opinion is that manuscript has sufficient quality to be published. On the other hand, I would like to remark these points for future work: from a formal view, in introduction are 10 sources, which is in my opinion a modest overview of the actual development and research in the area of inspection systems and Deep learning. From a practical view, The actual AI systems are relative complex to understand every step of working the system and know the edge conditions. My opinion is, the Deep learning methods are well explored and improved to suffiction defect detection according to trained dataset. And this way, the actual question of defect detection in research is suitable to explore: "Is it possible to detect the untrained or unknown defects for inspection system by conventional Deep learning models ?" (Lines 265-266). In the research, there are articles focusing on this issue, for instance: Autoencoder anomaly detection, Unsupervised learning, hybrid systems, and many others. And I would like to ask, if it would be possible in the future to provide open access database of defects and samples to the development inspection systems in this area? I think many researchers would be grateful for data from real industrial Environments. 

Again, thanks! And good luck in the job!

Have a nice day.

Reviewer.

Author Response

18 February 2022

Dear reviewer in Applied Sciences,

We sincerely thank you for your comments, “Smart System to Detect Painting Defects in Shipyards: Vision AI, a Deep-Learning Approach,” Manuscript ID applsci-1583982.

We acknowledge that the quality of our manuscript was improved by the comments of the reviewers and editors.

COMMENTS: Thanks to authors for answering my questions. First, I appreciate the deployment inspection system in the real Industry condition. My opinion is that manuscript has sufficient quality to be published. On the other hand, I would like to remark these points for future work: from a formal view, in introduction are 10 sources, which is in my opinion a modest overview of the actual development and research in the area of inspection systems and Deep learning. From a practical view, The actual AI systems are relative complex to understand every step of working the system and know the edge conditions. My opinion is, the Deep learning methods are well explored and improved to suffiction defect detection according to trained dataset. And this way, the actual question of defect detection in research is suitable to explore: "Is it possible to detect the untrained or unknown defects for inspection system by conventional Deep learning models ?" (Lines 265-266). In the research, there are articles focusing on this issue, for instance: Autoencoder anomaly detection, Unsupervised learning, hybrid systems, and many others. And I would like to ask, if it would be possible in the future to provide open access database of defects and samples to the development inspection systems in this area? I think many researchers would be grateful for data from real industrial Environments.

Response : We appreciate your comment and thank you for understanding the exact purpose of this paper and our effort. In particular, we strongly agree with the phrase "The actual AI systems are relational complex to understand every step of working the system and know the edge conditions."

The purpose of this paper is to share not only the AI research, but also the understanding of the production process, the understanding of why it occurs, and how to design and apply it. We are grateful for your understanding.

Regarding the disclosure of data, while we support the disclosure, internally, the company concluded that the data it to be limited to academic purposes under the NDA(Non-disclosure agreement).

This system is introduced as an item that provides DSME quality control and transparency, and reliability of the production process to ship owners. The company approved the preparation of this paper, but currently, disclosing raw data is not allowed with awareness of competitors.

However, we hope that this data will be released and used in more advanced applications in the future. For example, we hope that we will be able to obtain higher accuracy with various algorithms and methods that we have not thought of by opening the composition while disclosing it to Kaggle.

Internally, we are continuing to research ways to upgrade various algorithms with this data. We are also planning to develop autoAI such as autoML, which supports pipe-line construction and auto-labeling as well as the unsupervised-learning and hybrid method that commented.

As a study that applies data acquisition or AI to the field, a system that checks material through OCR and detects defects of steel surfaces in various shapes and situations using Deep Learning (similar issue has been disclosed in Kaggle competition; Severstal: Steel Defect Detection Can you detect and classify defects instel?) is also undergoing the process of acquiring and developing data, and these parts will also be disclosed in future papers.

Furthermore, we intend to be of great help to scholars who study deep learning by disclosing as much data as possible through continuous consultations within the company.

We believe that recent technological advances, including AI, have resulted from information sharing and disclosure. Open source and Open Library are big contributions towards allowing AI to enter a new phase.

We will continue to make efforts in the future to expand the system for acquiring a lot of data and applying AI at industrial sites, and to disclose such research and data.
